

# On the importance of multiple-component evaluation of spatial patterns for optimization of earth system models – A case study using mHM v5.6 at catchment scale.

Julian Koch[1], Mehmet C. Demirel[1], Simon Stisen[1]

[1]Department of Hydrology, Geological Survey of Denmark and Greenland, Copenhagen, 1350, Denmark

*Correspondence to*: Julian Koch (juko@geus.dk)

**Abstract.** The process of model evaluation is not only an integral part of model development and calibration but also of
paramount importance when communicating modelling results to the scientific community and stakeholders. The modelling
community has a large and well tested toolbox of metrics to evaluate temporal model performance. On the contrary, spatial
performance evaluation is not corresponding to the grand availability of spatial observations readily available and to the
sophisticate model codes simulating the spatial variability of complex earth system processes. This study makes a
contribution towards advancing spatial pattern oriented model calibration by rigorously testing a multiple-component
performance metric. The promoted SPAtial EFficiency (SPAEF) metric reflects three equally weighted components:
correlation, coefficient of variation and histogram overlap. This multiple-component approach is found to be advantageous
in order to achieve the complex task of comparing spatial patterns. SPAEF, its three components individually and two
alternative spatial performance metrics, i.e. connectivity analysis and fractions skill score, are applied in a spatial pattern
oriented model calibration of a catchment model in Denmark. Results suggest the importance of multiple-component
metrics, because stand-alone metrics tend to fail to provide holistic pattern information to the optimizer. The three SPAEF
components are found to be independent which allows them to complement each other in a meaningful way. In order to
optimally exploit spatial observations made available by remote sensing platforms this study suggests applying bias
insensitive metrics which further allow comparing variables which are related but may differ in unit. This study applies
SPAEF in the hydrological context using the mesoscale Hydrologic Model (mHM; version 5.6), but we see great potential
across disciplines related to spatial distributed earth system modelling.

## 1 Introduction

Spatially distributed models, which represent various components of the earth system, are extensively applied in policy-
making, management and research. Such modelling tackles a wide range of environmental problems, such as  the analysis of
drought patterns (Herrera-Estrada et al., 2017), assessing the spatial regularization of fertilizers in agricultural landscapes
(Refsgaard et al., 2014), modelling vegetation dynamics (Ruiz-Pérez et al., 2016) or forecasting spatial patterns of severe





weather under a changing climate (Gilleland et al., 2016). Earth system dynamics are typically characterized by a distinct spatial dimension which constitutes the mayor obstacle for many modelling efforts with respect to model structure, parametrization and forcing data.

In order to establish confidence in outputs generated by spatially explicit earth system models and further to justify their
application while recognizing their limitations it is of paramount importance to quantify performance (Alexandrov et al., 2011; Hagen and Martens, 2008; Kumar et al., 2012). Within the field of distributed hydrological modelling the call for a paradigm shift away from temporal model evaluation of aggregated variables such as discharge or hydraulic head towards a spatial pattern oriented model evaluation using independent spatial observations has been ongoing for nearly two decades (Grayson and Blöschl, 2001; Koch et al., 2016a; Stisen et al., 2011; Wealands et al., 2005). In fact modelling temporal
dynamics of hydrological response can be considered independent of a models spatial component as different parameters control these two dimensions of model performance (Pokhrel and Gupta, 2011). Along the lines of Gupta et al. (2008), the feasibility of an adequate spatial pattern oriented model evaluation is constrained by the versatility of the applied performance metric. The task to quantitatively compare spatial patterns is non-trivial and the multi-layered content of spatial patterns expresses distinct requirements to such a metric (Cloke and Pappenberger, 2008; Gilleland et al., 2009; Vereecken et
al., 2016). A single metric will generally not adequately address performance and instead a combination of metrics spanning over multiple relevant aspects of model performance are necessary (Clark et al., 2011; Gupta et al., 2012).

Model evaluation targeted at spatial performance requires reliable spatial observations which are broadly facilitated by remote sensing platforms across various spatial scales (McCabe et al., 2008; Orth et al., 2017). At small scale, Glaser et al. (2016) explored the applicability of portable thermal infrared cameras to evaluate simulated spatial patterns of surface
saturation in the hillslope-riparian-stream interface. At catchment scale, Schuurmans  et al. (2011) incorporate remote sensing based maps of latent heat in order to identify structural model deficiencies. At regional scale, Mendiguren et al. (2017) applied a spatial pattern oriented model evaluation based on remote sensing estimates of evapotranspiration to diagnose shortcomings of the national hydrological model of Denmark. At large scale, Koch et al. (2016b) utilized land surface temperature retrievals to evaluate large scale land surface models across the continental U.S..
The applicability of remote sensing data to calibrate hydrological models has already been explored by several studies that incorporated spatial patterns of land-surface temperature (Stisen et al., 2017), snow cover (Terink et al., 2015) or latent heat (Immerzeel and Droogers, 2008). Overall the merit of constraining model parameters against spatial observations has been widely recognized by the modelling community. However, the design of the performance metric which ensures that the spatial information, contained in the remote sensing data, is utilized optimally to inform the model calibration is rarely
touched upon in literature.

Bennett et al. (2013) provide an excellent overview of measures that allow the modeller to quantify performance of earth system models. They considered model evaluation a vital step during the iterative process of model development hence it can identify the need for additional data, alternative calibrations or updated model structure. This further emphasizes the need for robust performance metrics.





Our study highlights the development and application of a versatile metric that has the potential to advance the credibility of spatially distributed earth system models. When designing such a metric it is important to reflect on requirements as well as frameworks to properly test it in, which has been extensively discussed in literature (Cloke and Pappenberger, 2008; D. N. Moriasi et al., 2007; Dawson et al., 2007; Krause et al., 2005; Refsgaard and Henriksen, 2004; Schaefi and Gupta, 2007).

Following these references and our own reflections we identified the following five mayor requirements of a spatial performance metric: (1) The metric should be easy to compute, which makes results reproducible and creates credibility within the scientific community. (2) In order to be informative during model calibration the metric should be robust and deliver a continuous response to changes in parameter values. (3) In the formulation of the metric, multiple independent components are necessary to provide a holistic evaluation of the models performance. (4) The metric should offer the

possibility to compare related variables of different units; e.g. observed latent heat ($W/m^2$) and simulated evapotranspiration (mm/day). This enables evaluation via proxies and facilitates bias insensitivity which is found favourable, because it focuses on the pattern information contained in the remote sensing data instead of absolute values at grid scale. (5) The metric should be easy to communicate both inside and outside the scientific community. This requires a predefined range and the possibility to put metric scores into context; i.e. what value ensures satisfactory performance? Can we directly compare

scores between different catchments/models? These five points were carefully taken into consideration by Demirel et al (2017) for the formulation of SPAtial EFficiency (SPAEF) which they successfully applied in a spatial pattern oriented model calibration.

In this study, we rigorously test SPAEF and compare it with two additional spatial performance metrics; namely fractions skill score (Roberts and Lean, 2008) and connectivity analysis (Koch et al., 2016b). All three metrics are applied in a spatial

pattern oriented calibration of a catchment model using the multiscale Hydrologic Model (mHM: Samaniego et al., 2010). Such rigorous metric testing and comparison helps to generate familiarity and is inevitable in order to establish novel metrics in the scientific community.

## 2 Data and Methods

### 2.1 Study Site

The Skjern river catchment is located in the western part of the Danish peninsula. The catchments size amounts to 2500 km$^2$ and it has been studied intensively for almost a decade by the HOBE project (Jensen and Illangasekare, 2011). The climate is maritime with a mean annual precipitation of around 1050 mm which is partitioned in more or less equal amounts of streamflow and actual evapotranspiration. Topography slopes gently from the highest point of approximately 125 m elevation in the east to sea level in the western side of the catchment. Figure 1 shows the spatial variability of soil texture

which stresses that soils are predominately sandy with intertwined till and clay sections. Land use is dominated by arable land with patches of coniferous forest. The Skjern catchment does not exhibit a strong spatial gradient in hydrological response, because general gradients in catchment morphology or climatology do not exist. This promotes the catchment to be



an excellent test case for a spatial pattern oriented model calibration, because the simulated spatial patterns of hydrological variables are governed by optimizable parameters such as soil and vegetation properties.

## 2.2 Hydrological Model

This study utilizes the mesoscale Hydrologic Model (mHM; version 5.6) which is a grid based spatially distributed hydrological model (Kumar et al., 2013, 2010, Samaniego et al., 2010a, 2010b). The model accounts for key hydrological processes such as canopy interception, soil moisture dynamics, surface/subsurface flow generation, snow melting, evapotranspiration and others. Daily meteorological data forces the model and a gridded digital elevation model (DEM) characterizes the morphology of the catchment. Additionally, the spatial variability of observable physical properties such as soil texture, vegetation and geology are incorporated in the model structure as well. A multi-parameter regionalization technique enables mHM to consolidate three different spatial scales: meteorological forcing at coarse scale, intermediate model scale and fine scale morphological data. In case of the Skjern model, forcing data is available at 10-20 km resolution, the DEM is used at 250m scale and the model is executed at 1km scale. Effective parameters at the modelling scale are regionalized through nonlinear transfer functions which link spatially distributed basin characteristics at finer scale by means of global parameters which can be determined through calibration. Following the work presented by Demirel et al. (2017), the existing model structure was extended in order to adequately reflect the hydrological conditions of the Skjern river basin. This was achieved by adding effective calibration parameters to the soil moisture stress function, root fraction coefficient and the dynamic scaling of reference ET by incorporating the Moderate Resolution Imaging Spectroradiometer (MODIS) 8-day Leaf Area Index (LAI) product at 1 $km^2$ resolution. For further details we refer to the abovementioned reference.

## 2.3 Reference Data

The observational data which are employed as reference in the calibration is given in Figure 2 and consists of two datasets. First, 8 years (2001-2008) of discharge time series at two locations within the catchment where the first drains around 60% of the catchment area and the second an additional 25% (Figure 1). Second, in order to complement the temporal data we provide a remote sensing estimate of latent heat for cloud-free grids in June between 2001 and 2008. The month of June is the peak of the growing season which makes the spatial pattern distinct and relevant for a hydrological model evaluation. This reference spatial pattern is obtained by the Two Source Energy Balance Model (TSEB) (Norman et al., 1995). A detailed description of the remote sensing based estimation of latent heat across Denmark is presented by Mendiguren et al. (2017). As outlined by Mendiguren et al. (2017), TSEB represents a two layer model which separates soil and vegetation. Energy fluxes are estimated based on various input parameters and forcings among which land-surface-temperature (LST) and air temperature are found to be most sensitive. Input data for TSEB are obtained from the daytime LST MODIS product at 1 km spatial resolution. The reasoning behind averaging the latent heat maps in time to a mean monthly map is expressed two fold. First, daily spatial patterns are influenced by clouds and thus vary highly in coverage which limits the pattern information content. Second, daily estimates are associated with higher uncertainty and are more affected by forcing data;



e.g. the spatial distribution of precipitation on the previous day. Hence, aggregated monthly maps of latent heat represent a robust average that is more informative in a model calibration than daily maps, because it constitutes the imprint of soil properties and vegetation on the simulated pattern. Opposed to model forcing these are parameters that can actually be calibrated in a hydrological model.

## 5 2.4 Spatial Performance Metrics

### 2.4.1 Spatial Efficiency

For the formulation of a straightforward spatial performance metric we found inspiration in the Kling–Gupta efficiency (KGE; Kling and Gupta, 2009) which is a commonly used metric in hydrological modelling to evaluate discharge simulations. It is characterized by three equally weighted components i.e. correlation, variability and bias.

$$KGE = 1 - \sqrt{\left(\alpha_Q - 1\right)^2 + \left(\beta_Q - 1\right)^2 + \left(\gamma_Q - 1\right)^2} \qquad (3)$$

$\alpha_Q = \rho(obs, sim)$ and $\beta_Q = {\sigma_{sim}}/{\sigma_{sim}}$ and $\gamma_Q = \frac{(\mu_{sim} - \mu_{obs})}{\sigma_{obs}}$

where $\alpha_Q$ is the Pearson correlation coefficient between the observed (*obs*) and the simulated (*sim*) discharge time series, $\beta_Q$ is the relative variability based on the ratio of standard deviation in simulated and observed values and $\gamma_Q$ is the bias term which is normalized by the standard deviation of the observed data. KGE is selected as discharge objective function for the

optimization applied in this study.

The multiple-component nature of KGE is favourable, because a model evaluation can rarely be condensed to a single source of information. Instead a more holistic and balanced assessment using several aspects is favourable for a comprehensive model evaluation as advocated by Gupta et al. (2012), Krause et al. (2005) and others.

The difficulty to quantitatively compare spatial patters is illustrated in Figure 3 where two example patterns both generated

by mHM during calibration, are compared with the TSEB reference patter. A swift visual comparison clearly disambiguates that both are inadequate spatial pattern representations with respect to the reference; i.e. the first lacks spatial variability and the second miss spatial detail within the clearly separated clusters of high and low values. The Pearson correlation coefficient is a commonly known statistical measure that allows comparing two variables that are collocated in space and may differ in units. Despite the visual evaluation, both examples have a reasonably high correlation which allegedly suggests

good performance. When assessing the ratio of observed and simulated coefficient of variation it becomes clear that the first example lacks spatial variability whereas the distinct separation of the second example suggests an adequate representation of spatial variability. The deficiency of the second example becomes first clear when investigating the overlap of histograms of the normalized (z-score) of simulated and reference pattern. The z-score normalization results in a pattern with mean equal to zero and standard deviation equal to one, which is necessary to make two patterns with different units comparable.

The histogram match stresses non-existing spatial variability within the high and low areas, despite the satisfying correlation and spatial variability.





Based on the abovementioned examples and following the multiple-component idea of KGE we present a novel spatial performance metric denoted Spatial Efficiency (SPAEF) which was originally proposed by Demirel et al. (2017).

$$SPAEF = 1 - \sqrt{(\alpha - 1)^2 + (\beta - 1)^2 + (\gamma - 1)^2} \qquad (4)$$

$$\alpha = \rho(obs, sim) \text{ and } \beta = \frac{\left(\frac{\sigma_{sim}}{\mu_{sim}}\right)}{\left(\frac{\sigma_{obs}}{\mu_{obs}}\right)} \text{ and } \gamma = \frac{\sum_{j=1}^{n} min(K_j, L_j)}{\sum_{j=1}^{n} K_j}$$

where $\alpha$ is the Pearson correlation coefficient between the observed (*obs*) and simulated (*sim*) pattern, $\beta$ is the fraction of coefficient of variation representing spatial variability and $\gamma$ is the histogram intersection for the given histogram $K$ of the observed pattern and the histogram $L$ of the simulated pattern, each containing $n$ bins (Swain and Ballard, 1991). In order to enable the comparison of two variables with different units and to ensure bias insensitivity, the z-score of the patterns is used to compute $\gamma$. Throughout the manuscript $\alpha$ is referred to as *correlation*, $\beta$ as *cv ratio* and $\gamma$ as *histo match*.

**2.4.2 Connectivity**

The connectivity metric originates from the field of hydrogeology where it is commonly applied to characterise the spatial heterogeneity of aquifers (Koch et al., 2014; Rongier et al., 2016). Outside the hydrogeology community, connectivity analyses have also been conducted to describe spatial patterns of soil moisture (Grayson et al., 2002; Western et al., 2001) or land-surface temperature (Koch et al., 2016b). Following the classification of Renard and Allard (2013), the connectivity

analysis of a continuous variable is conducted via three steps: (1) a series of threshold percentiles decomposes the domain into a series of binary maps, (2) the binary maps undergo a cluster analysis that identifies spatially connected clusters and (3) the transition from many disconnected clusters to a single connected clusters can be quantified by principles of percolation theory (Hovadik and Larue, 2007). In this context the probability of connection ($\Gamma$) is considered a suitable percolation metric. $\Gamma$ states the proportion of pairs of cells that are connected among all possible pairs of connected cells of a cluster

map.

$$\Gamma(t) = \frac{1}{n_t^2} \sum_{i=1}^{N(X_t)} n_i^2, \qquad (5)$$

where $n_t$ is the total number of cells in the binary map $X_t$ below or above threshold $t$, which has $N(X_t)$ distinct clusters in total. $n_i$ is the number of cells in the $i$[th] cluster in $X_t$. The percolation is well captured by means of an increasing threshold that moves along all percentiles of the variable's range which makes this methodology bias insensitive. The connectivity analysis

is applied individually on cells that exceed a given threshold and those that fall below, which is referred to as low and high phase, respectively. Following Koch et al. (2016b), the root-mean–square-error between the connectivity at all percentiles of the observed ($\Gamma(t)_{obs}$) and the simulated ($\Gamma(t)_{sim}$) pattern denotes a tangible pattern similarity metric and can be calculated as:

$$RMSE_{Con} = \sqrt{\frac{\sum_{t=1}^{100} (\Gamma(t)_{obs} - \Gamma(t)_{sim})^2}{100}}. \qquad (6)$$





The average RMSE score of the low and the high phase is employed as the pattern similarity score for the connectivity analysis and is referred to as *connectivity* throughout the manuscript.

### 2.4.3 Fractions Skill Score

The fractions skill score (FSS) is a common metric in meteorology to provide a scale dependent measure that quantifies
spatial skill of various competing precipitation forecasts with respect to a reference (Mittermaier et al., 2013; Roberts and Lean, 2008; Wolff et al., 2014). In the FSS framework, a fraction reflects the occurrence of values exceeding a certain threshold at a given window size *n* and is calculated at each cell. Typically the thresholds are derived from the variable's percentiles, which constitutes the bias insensitivity of FSS (Roberts, 2008). The FSS workflow is defined by three main steps: (1) for each threshold, truncate the observed (*obs*) and the simulated (*sim*) spatial pattern into binary maps, (2) for each
cell, compute the fraction of cells that exceed the threshold and lie within a window of size *n\*n* and (3) calculate the mean-squared-error (MSE) between the observed and simulated fractions and normalize it with a worst case MSE ($MSE_{wc}$) that reflects the condition with zero agreement between the spatial patterns. The MSE is based on all cells ($N_{xy}$) that lie within the modelling domain with dimension of $N_x$ and $N_y$. For a certain threshold FSS at scale *n* is given by:

$$FSS_{(n)} = 1 - \frac{MSE_{(n)}}{MSE_{(n)wc}},$$   (7)

where

$$MSE_{(n)} = \frac{1}{N_{xy}} \sum_{i=1}^{N_x} \sum_{j=1}^{N_y} \left[ ref_{(n)ij} - scen_{(n)ij} \right]^2$$   (8)

and

$$MSE_{(n)wc} = \frac{1}{N_{xy}} \left[ \sum_{i=1}^{N_x} \sum_{j=1}^{N_y} ref_{(n)ij}^2 + \sum_{i=1}^{N_x} \sum_{j=1}^{N_y} scen_{(n)ij}^2 \right].$$   (9)

FSS ranges from zero to one, where one indicates a perfect match between *obs* and *sim* and zero reflects the worst possible
performance. For the simulated spatial patterns in the Skjern catchment we applied the concept of critical scales (Koch et al., 2017) and therefore selected three top and three bottom percentiles each assessed at an individual critical scale. The 1st, 5th and 20th percentiles focus on the bottom 1%, 5% and 20% of cells and are investigated at 25 km, 15 km and 5 km scale, respectively. Three top percentiles, 99th, 95th and 80th are analysed analogous. The average of the three top and bottom percentiles is calculated as an overall pattern similarity score and referred to as FSS throughout the manuscript.

### 2.5 Optimization Procedure

The mHM model of the Skjern catchment is applied at 1 km spatial resolution and the simulation period is set to 12 years (1997-2008) where the first four years are used as warm-up and the following eight years are utilized for the calibration. The model parameters are calibrated against observed discharge time series at two stations and the average latent heat pattern of June under cloud-free conditions. The reference pattern reflects an instantaneous observation of midday latten heat [w/m$^2$]
whereas the model simulates daily actual evapotranspiration [mm/day]. Obviously these variables are closely related;



however it requires suitable spatial performance metrics to be able to quantitatively compare two patterns with different units.

The sensitivity analysis of the 48 global parameters in the mHM setup for the Skjern catchment model was performed via two steps; a variance-based sequential screening (Cuntz et al., 2015) followed by a Latin-hypercube sampling (van Griensven et al., 2006). The first step identified 24 informative parameters and results were presented by Demirel et al. (2017). Subsequently we applied the Latin-hypercube sampling to further reduce the number of sensitive parameters to 17. Among the selected parameters, eight represent the soil moisture module (pedo transfer functions, root fraction distribution and soil moisture stress), two control the interflow, one affects the percolation, two are sensitive to the baseflow and four define the ET module via the dynamic scaling function using MODIS LAI.

In order to reflect on the ability of different spatial performance metrics to optimize the pattern performance of the distributed hydrological model applied in this study we have designed six calibrations. All commence with the same initial parameter set and include KGE at both discharge stations as temporal objective functions. Additionally each optimization features one of the promoted spatial performance metrics: (1) SPAEF, (2) *correlation*, (3) *cv ratio*, (4) *histo match*, (5) FSS and (6) *connectivity*. The metrics, *correlation*, *cv ratio* and *histo match*, represent the three SPAEF components. The spatial objective functions aim at optimizing the average ET pattern of June and are weighted five times higher than the discharge objective functions. We expect the capability of the model to optimize simulated time series of discharge to be more versatile in comparison to its flexibility to optimize spatial patterns which justifies the weighting of the objective functions. The optimizations were conducted with help of PEST (version 14.02) (Doherty, 2005) and the Shuffled Complex Evolution (SCE-UA) algorithm (Duan et al., 1993) was selected as optimizer. SCE-UA is considered a global optimizer and for our application it was set up to operate on two parallel complexes with 35 parameter sets in each complex. Each calibration was limited to 2500 model runs, which was found reasonable to allow convergence of the objective functions.

The simulation results from the initial parameter set are depicted in Figure 2. The simulated pattern of AET is almost uniform with very little spatial variability which results in a low SPAEF score of -0.58. The simulated discharge has the correct timing at both stations, where station #2 is clearly less biased than station #1. Both have reasonable KGE scores on the basis of the initial parameter set: 0.6 (station #1) and 0.7 (station #2).

## 3 Results and Discussion

### 3.1 Optimizing Spatial Patterns

Figure 4 visualizes the results from the six conducted calibrations with the aim to track the spatial patterns of simulated ET during the course of the optimization. SCE-UA is executed in iterative manner where each iteration reflects a shuffling loop in which a number of parameter sets are tested. In order to inter-compare the optimization progress across the six calibrations, Figure 4 illustrates the optimal spatial patterns at four selected iterations during the calibration. The second



iteration is the first where SCE-UA receives feedback from the applied metric after executing random sets of parameter values in the first iteration. Iterations 6 and 10 show intermediate steps from the optimization progress. The optimal spatial pattern depicts the final result in accordance to the six tested performance metrics after 2500 model runs.

From a metric point of view, the scores of the objective functions are improved for all six calibrations. Among the six
metrics, *connectivity* is the only one which has to be reduced to zero; the remaining metrics have an optimal score of one. The improvements from iteration 10 to the optimal parameter set are numerically marginal and visually not to be discriminated, which indicates convergence. The visual differences between the optimized spatial patterns are striking and the three metrics that consider local constrains (SPAEF, *correlation* and FSS) can clearly be distinguished from the remaining three. With respect to the reference pattern in Figure 2, the separation between forest and non-forest has been
inversed by optimizing against *cv ratio* and *connectivity*, because the right allocation is not reflected by the metrics. The *histo match* metric is based on z-score normalization which results in a clear underestimation of spatial variability.

The importance of human perception based model evaluation has been widely recognized in literature (Grayson et al., 2002; Hagen, 2003; Koch et al., 2015; Kuhnert et al., 2005). Following our visual evaluation we regard the SPAEF optimization as the one being most similar to the reference in Figure 2. The three SPAEF components lead to very diverging solutions and,
combined as SPAEF, the optimization yields a spatial pattern which adequately reflects the imprint of both, vegetation and soil on the simulated ET patterns. FSS as objective function performs almost equally satisfying and revisiting the defined critical scales may improve this calibration result even further.

All metrics contain different spatial information which is used to constrain the model parameters which results in optimized spatial patterns that clearly differ from one another. Although some metrics undoubtedly fail at informing the optimizer to
identify a parameter set satisfying our visual criterion they still provide relevant pattern information to a certain extent. In consequence, these metrics do not function as stand-alone objective functions for this calibration study; e.g. *cv ratio* yields an inadequate spatial pattern but as a component in SPAEF it generates an satisfying solution to the optimization problem. Following Krause et al. (2005), one should carefully take the pros and cons of each performance measure into consideration when designing the calibration/validation framework of a model. Moreover, the most suitable metric should be tailored to
the intended use of the model and should relate to simulated quantities which are deemed relevant for the application of the model. For the objective of our calibration study the bias insensitivity and the capability of a metric to compare variables that are related but differ in unit was most relevant.

Table 1 cross-checks the metrics scores of the six optimized spatial patterns in Figure 4. Reading the table column-wise allows investigating if the metrics provide independent information to the optimizer.  As an example, *cv ratio* reaches its
optimal score, however the reaming metrics perform poorly. This indicates that *cv ratio* conveys independent information with respect to the other metrics. On the other hand, calibrating against *correlation* yields a high FSS score which attests partly redundant information content in the two given metrics. Reading the table row-wise, screens for consistency of the calibrations. The highest metric score should be reached when calibrating against itself, which is the case for all six calibrations.



Additionally, Table 1 presents the KGE scores for the six conducted calibrations. The discharge performance has been improved by all calibrations and the scores vary slightly across them. Similar to the initial run station #2 performs generally better than station #1. The simulated discharge of the six optimized models is shown in Figure 5 for a 4 year period at station #1. All calibrations simulate the discharge dynamics in accordance to the observations and are generally equipped with a

good timing of the peak flows. Differences are to be found in the recession flow between the six simulations. However, our effort focuses on the spatial performance and it is striking how different the simulated spatial patterns can be while predicting almost identical streamflow. This supports previous findings in literature which stress that spatial and temporal response in hydrological models are controlled by different parameters and that the one cannot be used to inform the other (Pokhrel and Gupta, 2011; Stisen et al., 2011 and others).

Choosing a suitable metric alone is not sufficient to undertake a successful spatial pattern oriented model calibration. Model agility promoted by a flexible parametrization is required to allow the simulated spatial patterns to be optimized with respect to a reference pattern (Mendoza et al., 2015). In this study, this is achieved by applying a model code (mHM: Samaniego et al., 2010) that features regionalization techniques where spatially distributed basin characteristics are transformed via global parameters to effective model parameters at model scale. On the contrary, Corbari and Mancini (2014) conducted a spatial

validation of a subsurface – surface - land surface model against MODIS LST where parameters were calibrated individually at each grid. Opposed to the regionalization technique, this approach does not grand physically meaningful parameter fields and may overestimate the credibility of remote sensing data.

In order to further advance opportunities of spatial pattern oriented model evaluation, hydrological models can be extended by emission models to simulated brightness temperature which is closer to the true observations of the remote sensing

sensors. As an example, Schalgeet al. (2016) implemented such a coupling which facilitated the direct model evaluation against SMAP brightness temperature. Similar solutions are feasible for LST and it has the clear advantage of bypassing the uncertainties and inconsistencies associated to the remote sensing models which the hydrological modeller has no control of.

## 3.2 Spatial Efficiency Metric

Establishing novel metrics in the modelling community is often hindered by an intrinsic inertia supported by an excessive

choice of metrics which leads to reliance on familiar metrics. Both, the implementation and the interpretation of unfamiliar metrics may be found too troublesome by many users. Familiarity can only be obtained by rigorous testing and by having a metric which provides scores in a predefined range easy to interpret. In the following we will provide detailed analysis of the SPAEF calibration results to further understanding of its implications and the interaction between the three components.

Figure 6 depicts a 3-dimensional Pareto front of the three SPAEF components on the basis of the 2500 parameter sets

executed in the SPAEF calibration which allows investigating trade-offs between different objective functions. The formulation of SPAEF gives equal weights to the three components; hence the best compromise is the parameter set with the lowest Euclidian distance to the optimal point (1,1,1). If desirable, the weights could be adjusted manually to specifically



focus on one the three components. Throughout calibration, scores across the range of each component are obtained which indicates that the components are clearly sensitive to changes in spatial performance. Further it reveals the global nature of SCE-UA which rigorously explores the parameter space. With an ideal score of one, SCE-UA optimized SPAEF to 0.56, which may seem surprisingly low given the good visual agreement. This underlines that SPAEF is a tough criterion with

three independent components that individually penalize the overall similarity score. The question of what marks an acceptable and satisfying SPAEF score is hard to generalize and probably depends on the pattern to be assessed. The ET pattern in the Skjern catchment is dominated by local feedbacks of soil and vegetation, which constitute challenging small scale details to a model. Alternatively, a catchment with a strong spatial gradient of e.g. precipitation or topography may yield naturally a higher SPAEF score. Such gradients in forcing or morphology are typically not calibrated and will dominate

the spatial pattern of the estimated hydrological fluxes. A distinct spatial variability provided by the model inputs are therefore expected to favor *correlation* and *cv ratio* resulting in a higher SPAEF score. However more work is needed to study the relationship of spatial variability and SPAEF.

As introduced earlier, the human perception is considered a reliable benchmark for the evaluation of spatial performance metrics. More precisely, a metric can be regarded reliable if it is able to emulate the human vision. In order to establish a

reliable benchmark dataset, Koch and Stisen (2017) have conducted a citizen science project with the aim to quantify spatial similarity scores based on the human perception. Their study was based on over 6000 simulated spatial pattern comparisons of land-surface variables in the Skjern catchment. When being compared to the human perception SPAEF provides a satisfying coefficient of determination of 0.73. In comparison, the coefficient of determination for *connectivity*, FSS and *correlation* are 0.48, 0.60 and 0.76, respectively.

Figure 7 highlights the evolution of the three SPAEF components by tracking their scores during the 2500 runs of four calibrations: SPAEF, *correlation*, *cv ratio* and *histo match*. Convergence can be observed for all components when being calibrated against itself or SPAEF. This underlines that the choice to limit the optimizer to 2500 runs was reasonable. The results underline consistency, because SPAEF provides the second best score for all components right after being calibrated against itself. Furthermore, the three components can be considered independent, because optimizing against one component

does not automatically lead to improvement of another. This is especially the case for the *cv ratio* calibration where *correlation* stagnates and *histo match* decrease throughout the course of the 2500 runs. A weak relationship is found between *correlation* and *histo match*, which seems reasonable given the parametrization of the model. Further, both metrics are independent of spatial variability; however the right z-score distribution can only be achieved by correctly allocating low and high values, because the reference pattern exhibits a left skewed distribution (Figure 3).

**4 Conclusions**

The complexity of earth system models is currently increasing so does the availability of satellite based remote sensing observations. In light of the vast amount of already existing remote sensing products in combination with recent



developments, such as the promising Copernicus program with its multi-satellite Sentinel missions (McCabe et al., 2017), the incorporation of detailed spatial data retrieved from remote sensing platforms will continue to enable grand opportunities for earth system modelling in the near future.

This study aimed at making a contribution towards that course by rigorously testing SPAEF, a simple and novel spatial performance metric which has the potential to advance spatial pattern oriented validation and calibration of spatially distributed models. The applicability of SPAEF was tested in the hydrological context; however its versatility promotes it to be beneficial throughout many disciplines of earth system modelling.

We applied SPAEF alongside its three components and two other spatial performance metrics (*connectivity* and FSS) in a calibration experiment of a meso-scale catchment (~2500km$^2$) in Denmark. A satellite retrieved map of latent heat which represents the average evapotranspiration pattern of cloud free days in June was utilized besides discharge time series as the reference dataset. We draw the following main conclusions from this work:

- Quantifying spatial similarity is a non-trivial task and it requires taking several dimensions of spatial information simultaneously into consideration. The formulation of SPAEF is therefore based on three equally weighted components; i.e. correlation, ratio of coefficient of variation and z-score histogram overlap between a simulated and an observed pattern. SPAEF reflects the Euclidian distance of the three components from the optimum, which is equivalent to the concept of a three dimensional Pareto front. The components are bias insensitive and allow assessing two variables that differ in units. Further we could infer independent information content to the three components which complement each other when used jointly as SPAEF.

- SPAEF is straightforward to compute and has a predefined range between $-\infty$ and one which simplifies communication with the scientific community and stakeholders. Nevertheless, more rigorous testing is required to further establish familiarity. The relationship between SPAEF and spatial variability has to be investigated in more detail for the purpose of putting the metric into context; i.e. comparing different catchments or models.

- The right spatial performance metric alone is not enough to improve the spatial predictability of a distributed model trough calibration. The metric has to be accompanied by an agile model structure and flexible parametrization, such as regionalization techniques, allowing the simulated pattern to adjust in a meaningful way. Naturally, this has to be further supported by high quality forcing data, detailed catchment morphology and trustworthy spatial observations at adequate scale.

- The calibration exercise of the Skjern catchment highlighted the importance of incorporating spatial observation in the calibration of hydrological models since the six conducted calibrations yielded strikingly different ET patterns while simulating similar discharge dynamics. Based on our findings, bias insensitive spatial metrics are ideally accompanied by bias sensitive discharge metrics that secure the overall robustness in terms water balance closure.

With this contribution we hope to encourage the modelling community to rethink paradigms when formulating calibration/validation experiments by choosing appropriate metrics that focus on spatial patterns representing earth system processes.



**Code and Data Availability**

The code for the applied spatial performance metrics is made available via GitHub (https://github.com/JulKoch/SEEM). The mHM code is freely available on the UFZ homepage (http://www.ufz.de/mhm). All data used to produce the results of this paper will be provided upon request by contacting J. Koch.

**Acknowledgements**

The scientific work has been carried out under the SPACE (SPAtial Calibration and Evaluation in distributed hydrological modelling using satellite remote sensing data) project (grant VKR023443) which is funded by the Villum foundation.

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





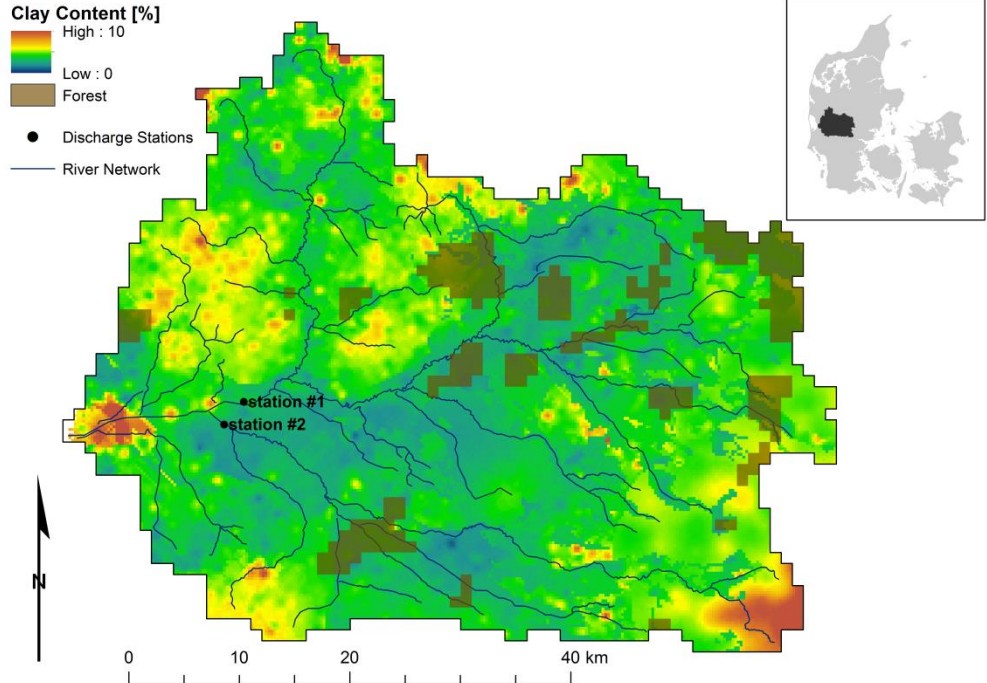

**Figure 1: Skjern river catchment in western Denmark. The map shows the spatial distribution of soil properties, forest areas and river network. Additionally, two discharge stations used in the optimizations are given.**





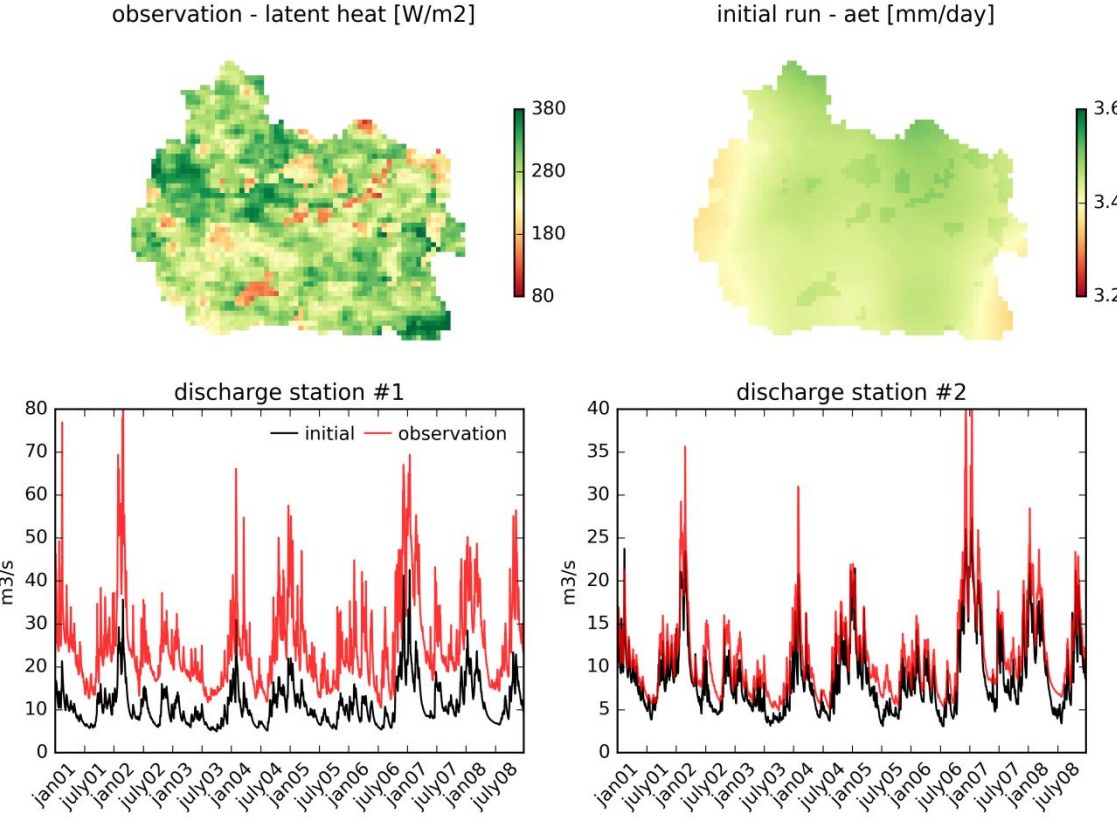

**Figure 2: Reference data used for the optimization: The average cloud free spatial pattern of midday latent heat in June (top left) and observed discharge (red line) at two stations (shown in Figure 1) for the 8 year simulation period (bottom). Also showing the simulation results from the initial parameter set: The average cloud free spatial pattern of daily actual evapotranspiration in June (top right) and the simulated discharge (black line) at the two reference stations.**



**Figure 3: Two examples to illustrate the importance of a multi-component analysis when comparing spatial patterns (top row). The maps are normalized by their mean. The histograms of the z-score normalized maps are presented in the middle row. The scatter plots of the mean normalized maps are given in the bottom row. Scores for the three SPAEF components (*histo match*, *cv ratio* and *correlation*) are given in the graphs.**





**Figure 4: Tracking of the simulated actual evapotranspiration maps (normalized by mean) throughout the six conducted optimizations using different objective functions. The first four columns show the trajectory of pattern improvements in accordance to one objective function. The maps depict the best fit between reference (bottom column) and model at various iterations throughout the optimization. The spatial similarity scores in accordance to the different metrics are given in the top-right corner of each map.**





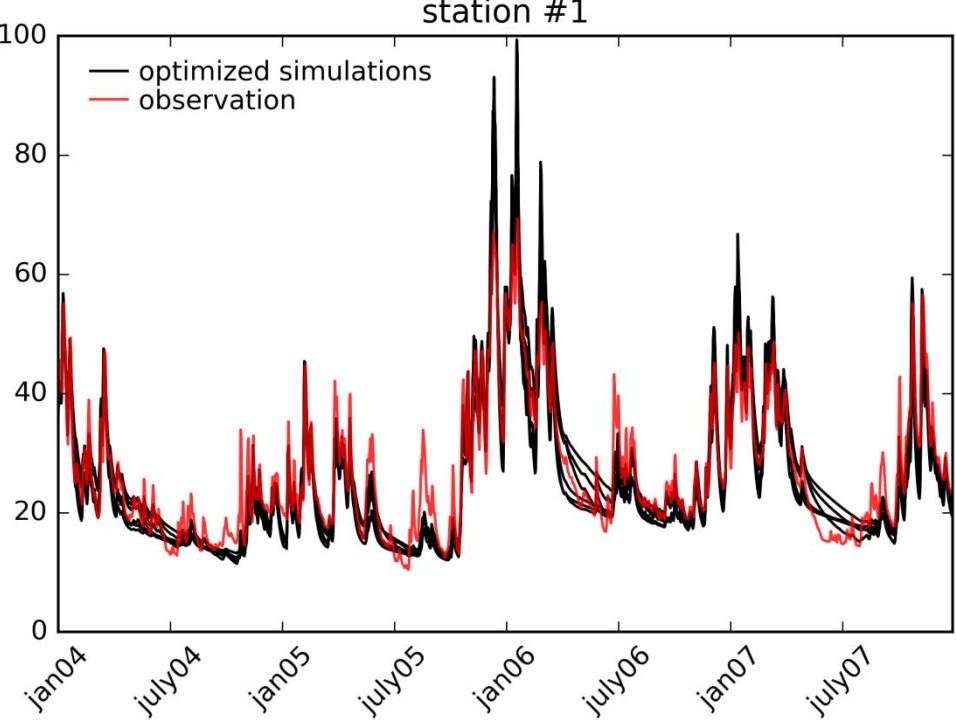

**Figure 5: Simulated discharge at station #1 obtained by the six optimizations. Data is shown only for four out of the eight years of simulation. KGE values vary between 0.84 and 0.95.**

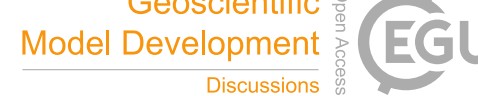



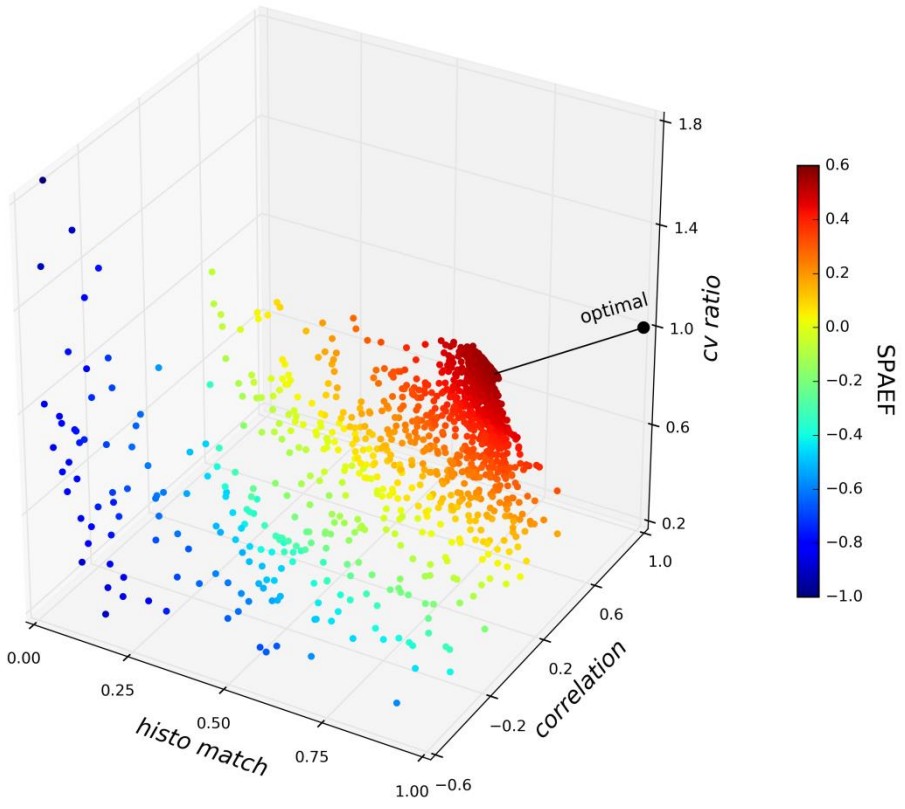

**Figure 6: 3D-pareto front based on the 2500 runs during the SPAEF optimization. Each component of the SPAEF metric represents an individual axis. The black line indicates the deviation between the theoretical optimal (1,1,1) SPAEF value and the optimized model run (0.72,0.73,0.81).**

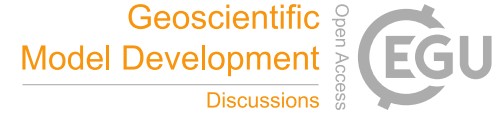



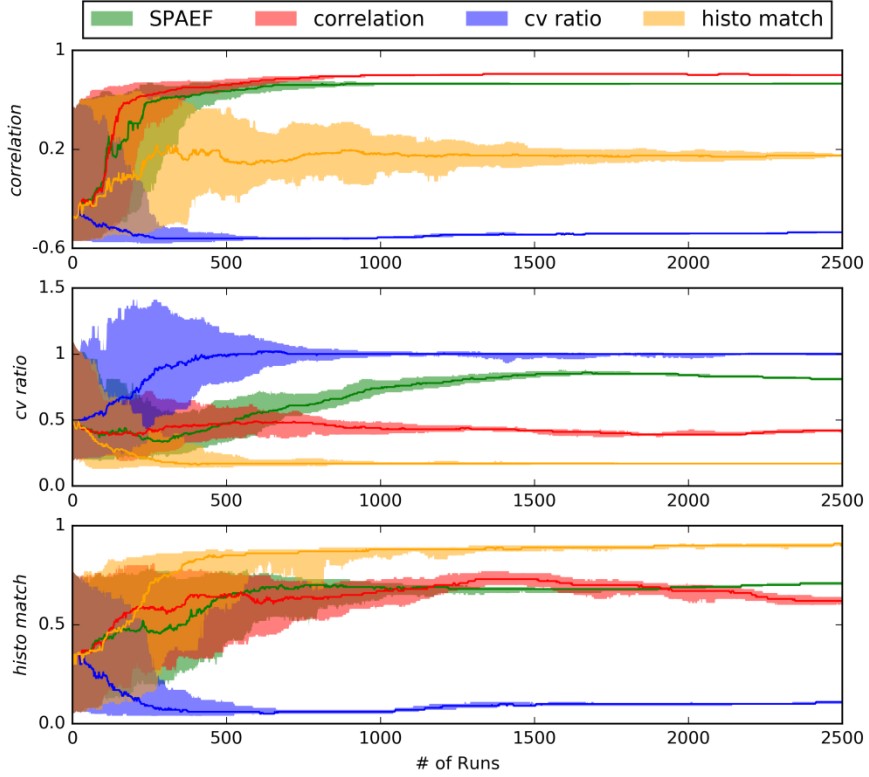

**Figure 7: Tracking of the three SPAEF components throughout the 2500 conducted runs of four calibrations (SPAEF, *correlation*, *cv ratio* and *histo match*). The envelopes represent the 10[th] and 90[th] percentile of a 100 run moving window; the line shows the median.**





**Table 1 Cross-check of the six conducted calibrations (as rows). The optimal model run is evaluated by the remaining metrics (as columns). Numbers in bold indicate the optimized value of the respective optimization.**

| Six optimizations | Calibrated against | | | | | |
|---|---|---|---|---|---|---|
| | SPAEF | *correlation* | *cv ratio* | *histo match* | *connectivity* | FSS |
| SPAEF | **0.56** | 0.28 | -0.74 | -0.19 | -1.16 | 0.18 |
| *correlation* | 0.73 | **0.80** | -0.48 | 0.15 | -0.56 | 0.74 |
| *cv ratio* | 0.81 | 0.41 | **1.00** | 0.17 | 2.17 | 0.57 |
| *histo match* | 0.72 | 0.64 | 0.10 | **0.91** | 0.08 | 0.36 |
| *connectivity* | 0.26 | 0.18 | 0.17 | 0.25 | **0.10** | 0.18 |
| FSS | 0.88 | 0.91 | 0.44 | 0.35 | 0.40 | **0.91** |
| KGE – station #1 | **0.89** | **0.90** | **0.88** | **0.84** | **0.88** | **0.95** |
| KGE – station #2 | **0.91** | **0.93** | **0.91** | **0.90** | **0.92** | **0.95** |

(Row group label: **Evaluated against**)