# Peer review of "The SPAtial Efficiency metric (SPAEF): Multiple-component evaluation of spatial patterns for optimization of hydrological models"

_Geoscientific Model Development, 2017_

## Short Comment (SC1) · 13 Dec 2017

GMD is encouraging authors to provide a persistent access to the exact version (??) of the source code used for the model version presented in the paper. As explained in https://www.geoscientific-model-development.net/about/manuscript_types.html the preferred reference to this release is through the use of a DOI which then can be cited in the paper. For projects in GitHub (such as SEEM) a DOI for a released code version can easily be created using Zenodo, see https://guides.github.com/activities/citable-code/ for details. For mHM you may consider to upload the program code of

the specifc version of the paper (including relevant data sets) as a supplement or make the code and data of the exact model version (v5.6) described in the paper accessible through a DOI (digital object identifier). In case your institution does not provide the possibility to make electronic data accessible through a DOI you may consider other providers (eg. zenodo.org of CERN) to create a DOI.

Please note that in the code accessibility section you can still point the reader to the GitHub repository for the newest version even if you use a DOI for the relevant releases.

Lutz Gross GMD Executive Editor

---

## Referee Comment (RC1) · Anonymous Referee #1 · 29 Dec 2017

**1 Summary**

The paper presents new metric that evaluates the spatial pattern of hydrologic model and earth system model. The new metric called SPAEF is multi-objectives, and consists of three components; spatial correlation, coefficient of variance ratio (simulation to observation), and histogram matching. The paper demonstrated mHM hydrologic model calibration by applying this metric to simulated ET distribution (or latent heat flux) against remote sensing data over 2500 sq-km catchment in Denmark and compared

the calibration performance against the use of the other metrics. The paper show that updated parameterization improves ET spatial pattern over use of the previous model parameters.

**2  Comments**

Goals of this paper, which is to propose new evaluation/calibration metric that quantifies the accuracy of spatial pattern of the earth system model, is good fit for GMD. Overall, I, as hydrologists who do modeling work, enjoyed reading the manuscript with great interest. My main comments below are regarding how this metrics and calibration strategy could be applied to the other model than mHMs, which might be hard to estimate spatially distributed parameters. My recommendation would be minor revision (if you can justify not performing additional simulations I mention in comment 4

1. To promote the metrics invented here, acronym of the metric is better pronounceable. Also, I would consider the metric name in Title. Just suggestion.

2. Please describe the weakness of two other metrics you evaluated besides SPAEF clearly.

3. The paper stated that spatial pattern of the model outputs depends at least on 1) process parameterizations (i.e., model equations), 2) accuracy of climate forcing (spatio-temporal pattern), and 3) parameter regionalization scheme (how parameters are distributed in space). I agree with these, but I speculate that spatial pattern is regulated in the first order by transfer function forms that convert soil/vegetation data to parameter values. Maybe mention this?

4. While mHM has a very unique regionalization scheme called mulit-scale parameter regionalization scheme (calibrate the coefficients of transfer functions that

compute parameter values from distributed geophysical data), making it easy to regionalize the parameters at any scales, all most all the other models do not have such a scheme. Therefore, it seems to be difficult to perform distributed model calibration presented in this paper for the other models. How applicable is this calibration strategy to the other models?

5. However, I still think this is an unique calibration strategy that combines spatial pattern and temporal pattern metrics, but meantime, I thought there need for more calibration experiments to understand the values of spatial pattern metrics for calibration purpose. I wish that there would have been results from 1) streamflow only calibration and 2) spatial pattern metric only calibration, showing skills of both ET spatial pattern and streamflow simulation. This way, the paper could show real value of this spatial pattern calibration. Does streamflow only calibration produce worse ET spatial pattern than the streamflow and ET combined calibration? Does spatial pattern only calibration produce worse streamflow simulations than the case streamflow is not used for calibration?

6. Contrast to hydrologic models, earth system model community do not have calibrate the parameters though Land surface model community started to pay more attention to calibrations/sensitivity analysis. Therefore, the presentation of this paper is more related to hydrologic model application. However, spatial pattern metrics could be used for model evaluation purpose. For example, would it be possible (or worthwhile) to use this for evaluation of meteorological fields from climate models against observation or reanalysis grid.

**3   Minor comments or specific line by line comments**

- I found a few typos – mayor-> major (P2, Line 2), patter->pattern (P5, Line 20).

- P5, Line3-4. I am not sure if I understand this sentence. Do you mean soil/vegetation properties by "these".

- P5. Q in KGE equation is incorrect. It should be $\mu_{sim}/\mu_{obs}$. Also, correct explanation in Line 14.

- P6, Line1-9. I think this paragraph is better fit after P5, L18.

- P9, Line6-7. Use of spatial pattern metrics as objective function converge faster than streamflow derived objective function. That seems to make sense because spatial pattern is by large determined by fixed transfer function forms and soil/vegetation properties in the mHM. It would be nice to mention the reason if you know.

- P10, Line10-14. I think this is good points to discuss, but I think it would be nice to discuss constrains from transfer function form (regularization equations).

- P11. Line 22. This number of iteration for convergences should depend on model choice and also regionalization scheme. So it is better not to generalize the conclusion here (I think).

- P11. Line26. I don't understand why it is reasonable given the parameterization of the mHM? Please elaborate a little more.

---

## Referee Comment (RC2) · Anonymous Referee #2 · 2 Jan 2018

The manuscript by Koch et al. proposes a multicomponent metric for evaluation and optimization of a hydrological model which can be used for any spatial pattern comparison. The topic is of interest for GMD and the manuscript is well structured, the conclusions well supported by adequate figures. I have no major concerns about the manuscript but a couple of suggestions that may help to improve the manuscript.

The two major comments are:

1) Title: The title emphasizes that it is a method for Earth system models. While the

manuscript strongly focusses on hydrological models. I am not a hydrologist and I found the Introduction too focussed on hydrological models and not very interesting for Earth system modellers. The title suggests a stronger overall discussion of Earth system models, while the whole paper is mainly about hydrological models, in the introduction as well as in the discussion. I suggest to remove the reference to Earth system models in the title to not raise wrong expectations.

2) your manuscript does not mention data uncertainty, while this could/should be a major component of a comparison metric too. if the model is within the uncertainty of observations further optimization would be overfitting. As more and more datasets provide data uncertainties, the possibility to include this information can be a major advantage over other metrics.

Specific comments:

There are a number of grammar and spelling errors throughout the manuscript. As Copernicus offers an editing service I do not detail these errors here.

p.1 l. 20: " to the optimizer", the optimization issue was not introduced before and is not relevant here. stand-alone metrics do not only fail to provide the necessary information to optimizers, but also an evaluation or calibration can suffer from only one quantified characteristic.

p.2 l. 1-3: I don't understand, earth system models usually have 2 spatial dimensions, but I dont see why they are and obstacle for modelling efforts. Do you mean the spatial scale or resolution? Even then I am not sure whether this is the major obstacle in general. Maybe it is for hydrological models? Otherwise please add a reference. It does not get clear from this sentence why this should be the case.

p.2. l. 6-9. These developments could be interesting if you would give more detail. It would also put your work better in the context. Do these approaches already use multicomponent metrics? what are the differences between the approaches of spatial

pattern oriented model evaluation? These examples are all from the field of hydrology? No other field of research has been dealing with such metrics?

p.2 l. 9-11: Strange. In Earth system modelling spatial and temporal scales are quite related. For instance the necessary temporal time step depends on the spatial resolution. also parameterizations might require adjustments due to changes in temporal or spatial resolutions. Maybe this is very specific for hydrological models?

p.2 l. 15-16: It might depend on the application of the model, sometimes the spatial pattern might even be irrelevant and a good temporal performance is sufficient. At some later point you mention that the necessary performance depends on the application of the model, but it might be useful to mention this already earlier in the introduction.

p.3 l. 1-5: are the requirements for earth system models and hydrological models the same? you claim your studies findings are imporant for earth system models but all your requirements and testing seem very focussed on hydrological models.

p.3 l. 9-12: if your variable has different units, ok. but if the unit is the same you might want your model to have the same mean or at least not a large deviation. That would then require an additional metric? how would you merge it then with your multicomponent metric?

p.3 l. 15: the possibility to include data uncertainties could be another point. remote sensing data inlude considerable uncertainties, optimizing the model by treating the "observed data" as the truth can lead to overfitting or biased model parameters especially if the uncertainties in the data scale with another important variable or increase with increasing values of the variable.

p.4 l.30: this seems your way to partly deal with the data uncertainty.

p.5, l.17, "source of information" this seems to be the wrong expression, probably a single metric or a single characteristic? single source of information sounds to me like using only one dataset to compare the model with as opposed to using multiple

datastreams to optimize or evalute the model.

p.6, l.3: why are you doing a sensitivity analysis? Is this to select a limited set of parameters for the optimization? if yes please explain.

p.6 l. 22-25: This seems to be a result, please move this paragraph.

p. 12. l. 14: The insensitivity to bias can also be a disadvantage, in many cases the optimized model is desired to be unbiased. p.12, l. 15: if the units differ, it might depend how the two units relate to each to other. it certainly is ok if they linearly scale. How about a nonlinear relationship? How about a possible change in sign as for instance with celsius and kelvin? if the mean temperature in celsius would go towards zero you would get difficulties for the beta part of your metric?

Reproduceability: Will you provide your model outputs, observations used and analysis scripts?

---

## Author Comment (AC1) · 25 Jan 2018

L. Gross

l.gross@uq.edu.au

GMD is encouraging authors to provide a persistent access to the exact version (??) of the source code used for the model version presented in the paper. As explained in https://www.geoscientific-model-development.net/about/manuscript_types.html the preferred reference to this release is through the use of a DOI which then can be cited in the paper. For projects in GitHub (such as SEEM) a DOI for a released code version can easily be created using Zenodo, see https://guides.github.com/activities/ citable-code/ for details. For mHM you may consider to upload the program code of the specifc version of the paper (including relevant data sets) as a supplement or make the code and data of the exact model version (v5.6) described in the paper accessible through a DOI (digital object identifier). In case your institution does not provide the possibility to make electronic data accessible through a DOI you may consider other providers (eg. zenodo.org of CERN) to create a DOI. Please note that in the code accessibility section you can still point the reader to the GitHub repository for the newest version even if you use a DOI for the relevant releases.
Lutz Gross GMD Executive Editor

*We would like to thank Executive Editor Lutz Gross for his comment. We will follow his suggestions and provide citable versions of the code used in our study. The hydrological model (mHM) is citable via Zenodo (10.5281/zenodo.1069202 ). All model code modifications used for this study, as described in detail by Demirel et al. (2018), are included in the recent mHM release (v.5.8). For the revision, we will update the version number of mHM respectively. The scripts for FSS and connectivity analysis are available in the SEEM repository on GitHub, which has been made citable using Zenodo (10.5281/zenodo.1154614). The code of the SPAEF metric is citable via ResearchGate using the following doi: 10.13140/RG.2.2.18400.58884. The exact version of the scripts including their DOIs will be provided in the revision. Forcing data and mHM parameter files will be made available upon request which will be clearly stated in the "Code and Data Availability" section in the revised manuscript. However, the DMI (Danish Meteorological Institute) forcing data can only be shared for pure research purposes and are available on the HOBE database (http://www.hobecenter.dk/index.php/data). The database requires a login, which can be obtained from the admin.*

*Demirel, M. C., Mai, J., Mendiguren, G., Koch, J., Samaniego, L., and Stisen, S.: Combining satellite data and appropriate objective functions for improved spatial pattern performance of a distributed hydrologic model, accepted for publication in Hydrol. Earth Syst. Sci., https://doi.org/10.5194/hess-2017-570, 2018.*

---

## Author Comment (AC2) · 25 Jan 2018

**1 Summary**

The paper presents new metric that evaluates the spatial pattern of hydrologic model and earth system model. The new metric called SPAEF is multi-objectives, and consists of three components; spatial correlation, coefficient of variance ratio (simulation to observation), and histogram matching. The paper demonstrated mHM hydrologic model calibration by applying this metric to simulated ET distribution (or latent heat flux) against remote sensing data over 2500 sq-km catchment in Denmark and compared the calibration performance against the use of the other metrics. The paper show that updated parameterization improves ET spatial pattern over use of the previous model parameters.

*We would like to thank the reviewer for his/her thorough revision of our manuscript. We are very pleased that our work on spatial pattern oriented model evaluation is generally well received by the reviewer. The comments raised by the reviewer pose valuable thoughts and the rigorous revision following his/her suggestions will certainly improve the scientific quality of our work. Our replies below indicate what we intend to change in the manuscript prior to resubmission.*

**2 Comments**

Goals of this paper, which is to propose new evaluation/calibration metric that quantifies the accuracy of spatial pattern of the earth system model, is good fit for GMD. Overall, I, as hydrologists who do modeling work, enjoyed reading the manuscript with great interest. My main comments below are regarding how this metrics and calibration strategy could be applied to the other model than mHMs, which might be hard to estimate spatially distributed parameters. My recommendation would be minor revision (if you can justify not performing additional simulations I mention in comment 4

1. To promote the metrics invented here, acronym of the metric is better pronounceable. Also, I would consider the metric name in Title. Just suggestion.

*We will follow the reviewer's advice and add the name of the metric including acronym in the title: "The SPAtial EFficiency metric (SPAEF): Multiple-component evaluation of spatial patterns for optimization of hydrological models". We agree that "SPAEF" may not be easy to pronounce, but this is nothing we have considered during the formulation of the metric. Also other popular metrics such as KGE or NSE are also not easily pronounceable*

2. Please describe the weakness of two other metrics you evaluated besides SPAEF clearly.

*We will add a clear discussion of the differences between SPAEF and Connectivity and FSS in the revised manuscript. Figure 4 as well as Table 1 can be used as illustrations to elaborate on the differences between the metrics. In comparison to SPAEF, Connectivity does not consider variability or the correct allocation. FSS constrains the right allocation but also does not explicitly handle variability. However, it may not be a completely fair comparison, because we argue that multiple components have to be taken into consideration when comparing spatial patterns. FSS and Connectivity have their strengths, but are single component metrics which perform less satisfactory in comparison to SPAEF. SPAEF is a multiple component metric which marks the key advantage over the other two metrics.*

3. The paper stated that spatial pattern of the model outputs depends at least on 1) process parameterizations (i.e., model equations), 2) accuracy of climate forcing (spatio-temporal pattern), and 3) parameter regionalization scheme (how parameters are distributed in space). I agree with these, but I speculate that spatial pattern is regulated in the first order by transfer function forms that convert soil/vegetation data to parameter values. Maybe mention this?

*We agree with the reviewer on this point, the transfer functions were the key element that allowed us to obtain such a satisfying result in terms of spatial pattern performance. However, the remaining two points are still relevant. The catchment used for this study is characterized by quite homogeneous climatic forcing and the monthly maps of ET are therefore less effected by climate in comparison to soil and vegetation. The spatial pattern calibration of a catchment with a strong climate gradient may be more constrained by the quality of the climate forcing than the Skjern catchment. Lastly, having the right process descriptions is essential to predict any physical system. We will make sure to point out the importance of the transfer functions in the revised manuscript.*

4. While mHM has a very unique regionalization scheme called mulit-scale parameter regionalization scheme (calibrate the coefficients of transfer functions that compute parameter values from distributed geophysical data), making it easy to regionalize the parameters at any scales, all most all the other models do not have such a scheme. Therefore, it seems to be difficult to perform distributed model calibration presented in this paper for the other models. How applicable is this calibration strategy to the other models?

*This is right, MPR allows easy regionalization in mHM, but its application is not limited to mHM. MPR can also be added to other model structures, as presented by Samaniego et al. (2017) for PCR-GLOBWB and Mizukami et al. (2017) for VIC. Samaniego et al. (2017) have outlined a modelling protocol to describe how MPR can be added to a particular model, which extends the applicability of MPR beyond mHM. We will provide the two references below and a discussion of the transferability of MPR to other models in the revised manuscript. Besides MPR, which is one way to implement parameter regionalization, in the calibration of distributed models, every modeler should think of way to regionalize parameters during calibration. This can be by self-implemented transfer functions which are added as a pre-processing script to the calibration routine. Regionalization is certainly not limited to MPR and simpler solutions may be sufficient in some cases to give the parameter fields the desired freedom to adjust a simulated to an observed spatial pattern.*

*Mizukami, N., Clark, M. P., Newman, A. J., Wood, A. W., Gutmann, E. D., Nijssen, B., Rakovec, O. and Samaniego, L.: Towards seamless large-domain parameter estimation for*

*hydrologic models, Water Resour. Res., 53(9), 8020–8040, doi:10.1002/2017WR020401, 2017.*

*Samaniego, L., Kumar, R., Thober, S., Rakovec, O., Zink, M., Wanders, N., Eisner, S., Müller Schmied, H., Sutanudjaja, E., Warrach-Sagi, K. and Attinger, S.: Toward seamless hydrologic predictions across spatial scales, Hydrol. Earth Syst. Sci., 21(9), 4323–4346, doi:10.5194/hess-21-4323-2017, 2017.*

5. However, I still think this is an unique calibration strategy that combines spatial pattern and temporal pattern metrics, but meantime, I thought there need for more calibration experiments to understand the values of spatial pattern metrics for calibration purpose. I wish that there would have been results from 1) stream-flow only calibration and 2) spatial pattern metric only calibration, showing skills of both ET spatial pattern and streamflow simulation. This way, the paper could show real value of this spatial pattern calibration. Does streamflow only calibration produce worse ET spatial pattern than the streamflow and ET combined calibration? Does spatial pattern only calibration produce worse streamflow simulations than the case streamflow is not used for calibration?

*The reviewer touches upon a very interesting point. Here we would like to refer to Demirel et al. (2017) who have conducted the above mention calibration experiments for the same model setup. They tested three calibration strategies: A calibration ensemble of Q-only, Spatial-only and a combination of Q and Spatial. Their findings underline the strength of combining temporal and spatial observations, as the uncertainty of predicting Q for the combined calibration was lower than the Q-only calibration. On the other hand, it was not possible for the Spatial-only calibration to constrain the hydrograph in a meaningful way. With respect to the spatial pattern performance, the Q-only calibration resulted in poor spatial patterns while very limited tradeoffs were noticeable comparing the spatial pattern performance of Spatial-only and the combined calibration. This underlines the limited trade-off between Q dynamics and spatial patterns illustrating the benefit of combining observation types in a multi-objective framework. We will refer to the results by Demirel et al. (2017) in the revised manuscript in detail to make the reader aware of the limited tradeoffs between temporal and spatial observations and the fact that spatial patterns have the power to constrain the hydrograph simulation efficiently when being paired with Q observations in a multi-objective calibration framework.*

*Demirel, M. C., Mai, J., Mendiguren, G., Koch, J., Samaniego, L. and Stisen, S.: Combining satellite data and appropriate objective functions for improved spatial pattern performance of a distributed hydrologic model, Hydrol. Earth Syst. Sci. Discuss., 1–22, doi:10.5194/hess-2017-570, 2017.*

6. Contrast to hydrologic models, earth system model community do not have calibrate the parameters though Land surface model community started to pay more attention to calibrations/sensitivity analysis. Therefore, the presentation of this paper is more related to hydrologic model application. However, spatial pattern metrics could be used for model evaluation purpose. For example, would it be possible (or worthwhile) to use this for evaluation of meteorological fields from climate models against observation or reanalysis grid.

*We completely agree to this point which has also been pointed out by reviewer 2. We decided to remove the emphasize on earth system models in the title and introduction and rather focus on the applicability of SPAEF for hydrological models. We will follow the suggestion of the*

*reviewer and add references which promote the usability of spatial pattern metrics to evaluate spatial patterns of metrological or atmospherical models*

3 Minor comments or specific line by line comments
• I found a few typos – mayor-> major (P2, Line 2), patter->pattern (P5, Line 20).

*Thanks, these will be corrected.*

• P5, Line3-4.I am not sure if I understand this sentence. Do you mean soil/vegetation properties by "these"?

*Exactly, we will change the sentence and try to be more specific.*

• P5. Q in KGE equation is incorrect. It should be $\mu_{sim}/\mu_{obs}$. Also, correct explanation in Line 14.

*Correct. We will update the bias terms in equation 3.*

• P6, Line1-9. I think this paragraph is better fit after P5, L18.

*Agree. We will reorder this section.*

• P9, Line6-7. Use of spatial pattern metrics as objective function converge faster than streamflow derived objective function. That seems to make sense be- cause spatial pattern is by large determined by fixed transfer function forms and soil/vegetation properties in the mHM. It would be nice to mention the reason if you know.

*We actually do not compare convergence rates between spatial and temporal objective functions, because we do not show any results that could support such a conclusion. Based on our results we comment on the convergence of the spatial objective functions which support our number of maximum runs for the calibration.*

• P10, Line10-14. I think this is good points to discuss, but I think it would be nice to discuss constrains from transfer function form (regularization equations).

*We will add a few points on the limitations of MPR, such as that the selection and definition of robust transfer functions can be difficult and bears uncertainties. Reliable transfer functions are crucial for the applicability of MPR. Other limitations are that the transfer functions are tedious to implement in other models besides mHM, as discussed above. Also, the minimum scale at which a model can be applied is depending on the data availability, since the subgrid variability is fundamental to MPR. The abovementioned limitations, among others, are discussed by Samaniego et al. (2017).*

*Samaniego, L., Kumar, R., Thober, S., Rakovec, O., Zink, M., Wanders, N., Eisner, S., Müller Schmied, H., Sutanudjaja, E., Warrach-Sagi, K. and Attinger, S.: Toward seamless hydrologic predictions across spatial scales, Hydrol. Earth Syst. Sci., 21(9), 4323–4346, doi:10.5194/hess-21-4323-2017, 2017.*

• P11. Line 22. This number of iteration for convergences should depend on model choice and also regionalization scheme. So it is better not to generalize the conclusion here (I think).

*Yes, we will down tone this conclusion and clearly state that this may only be relevant for our study.*

• P11. Line26. I don't understand why it is reasonable given the parameterization of the mHM? Please elaborate a little more.

*The relationship between histo match and correlation seems reasonable because of the slightly skewed distribution of the ET pattern (Figure 3). The lower side of the distribution are the forest grids, which have a lower ET during the growing season than the agricultural areas. Calibrating against histo match with such a peculiar distribution will result in a reasonable correlation, because low and high values will automatically be allocated correctly. This finding does not result in a crucial conclusion and it is further very much limited to this study and to the applied reference pattern. Therefore we will consider omitting these sentences in the revised version of the manuscript.*

---

## Author Comment (AC3) · 25 Jan 2018

The manuscript by Koch et al. proposes a multicomponent metric for evaluation and optimization of a hydrological model which can be used for any spatial pattern comparison. The topic is of interest for GMD and the manuscript is well structured, the conclusions well supported by adequate figures. I have no major concerns about the manuscript but a couple of suggestions that may help to improve the manuscript.

*We would like to thank the reviewer for his/her elaborated review of our manuscript. Overall, we are very pleased that our efforts to promote and evaluate the SPAEF metric are generally well received by the reviewer. We will follow the suggestions made by the reviewer to revise our manuscript and believe that it will strengthen the scientific quality of our work. Our replies below indicate what we intend to change in the manuscript prior to resubmission.*

The two major comments are:
1) Title: The title emphasizes that it is a method for Earth system models. While the manuscript strongly focusses on hydrological models. I am not a hydrologist and I found the Introduction too focussed on hydrological models and not very interesting for Earth system modellers. The title suggests a stronger overall discussion of Earth system models, while the whole paper is mainly about hydrological models, in the introduction as well as in the discussion. I suggest to remove the reference to Earth system models in the title to not raise wrong expectations.

*We totally agree to that point. Our original idea was to promote SPAEF to a broader audience since GMD covers earth modelling disciplines beyond hydrology. However we agree to the reviewer that our work is limited to the modeling of hydrological systems which we will clearly state in the revised manuscript. Other disciplines of earth system modelling may also work with spatially distributed models, but the way these models are parametrized and calibrated may differ from the hydrological community. This should also be reflected in the introduction of the revised manuscript. We also intend to change the title to: "The SPAtial EFficiency metric (SPAEF): Multiple-component evaluation of spatial patterns for optimization of hydrological models"*

2) your manuscript does not mention data uncertainty, while this could/should be a major component of a comparison metric too. if the model is within the uncertainty of observations further optimization would be overfitting. As more and more datasets provide data uncertainties, the possibility to include this information can be a major advantage over other metrics.

*We agree that data uncertainty should be an elementary consideration when evaluating models and that a metric should ideally reflect this. We have decided to deal implicitly with*

*data uncertainty in our study. This was achieved twofold, first through temporal aggregation to monthly maps of evapotranspiration and secondly through the bias insensitivity of the promoted metric. The temporal aggregation will remove noise and uncertainties in the observations may cancel out. Monthly maps of ET will be less affected by uncertain rainfall variability and the dominant pattern influenced by soil and vegetation will become more apparent. The fact that SPAEF is bias insensitive will also alleviate the effect of uncertainties in the observations. In the end, we do not assess the exact values at grid scale, instead we investigate global characteristics such as distribution and variability which are expected not to be strongly affected by data uncertainty. The correlation coefficient is part of the SPAEF formulation as well and may be more prone to data uncertainty, but again, we investigate the overall allocation of high and low values which will control the correlation and uncertainty is likely not to have a strong effect. These thoughts on data uncertainty will be added to the revised discussion section.*

Specific comments:

There are a number of grammar and spelling errors throughout the manuscript. As Copernicus offers an editing service I do not detail these errors here.

*We will pay special attention to detect any grammar and spelling errors during the revision of the manuscript. The remaining ones are then hopefully corrected by the editing team.*

p.1 l. 20: " to the optimizer", the optimization issue was not introduced before and is not relevant here. stand-alone metrics do not only fail to provide the necessary information to optimizers, but also an evaluation or calibration can suffer from only one quantified characteristic.

*We agree and will remove the reference "to the optimizer".*

p.2 l. 1-3: I don't understand, earth system models usually have 2 spatial dimensions, but I dont see why they are and obstacle for modelling efforts. Do you mean the spatial scale or resolution? Even then I am not sure whether this is the major obstacle in general. Maybe it is for hydrological models? Otherwise please add a reference. It does not get clear from this sentence why this should be the case.

*With the expression "spatial dimension" of earth system models we intendent to refer to the spatial variability. We agree that this may be confusing to the reader and will change it to the term "spatial variability".*

p.2. l. 6-9. These developments could be interesting if you would give more detail. It would also put your work better in the context. Do these approaches already use multicomponent metrics? what are the differences between the approaches of spatial pattern oriented model evaluation? These examples are all from the field of hydrology? No other field of research has been dealing with such metrics?

*We will elaborate more on the cited literature. The main point is that several other studies have highlighted the value of spatial observations in the evaluation of distributed models, but the main idea, to use multiple-components, has not been clearly addressed before. This marks the key novelty of our work and we will make sure that this is stated clearly in the revised version of our manuscript. We will extend the citing literature with examples outside the field of hydrology.*

p.2 l. 9-11: Strange. In Earth system modelling spatial and temporal scales are quite related. For instance the necessary temporal time step depends on the spatial resolution. also parameterizations might require adjustments due to changes in temporal or spatial resolutions. Maybe this is very specific for hydrological models?

*There may be a misunderstanding, we do not intend to refer to spatial and temporal scales. Instead we refer to spatial and temporal processes. The term "dimension" may be misleading and we will replace it with "variability". We want to point out that different parameters control temporal and spatial variability. The reviewer may be right that this a phenomena limited to the context of hydrological modelling, where our main expertise lies. In our experience, it is a challenging exercise to try to infer a meaningful spatial distribution of parameters by calibrating a hydrological model only against streamflow observations. The problem of equifinality arises where many parameter fields yield the same hydrograph. On the other hand, calibrating a model against spatial patterns only does not necessarily yield a meaningful hydrograph. This issue was addressed by Demirel et al., 2017 who applied the same model setup to conduct several calibration experiments: One using only streamflow data and another using only spatial patterns. This study highlighted the independency of the temporal and spatial observations and, when used jointly in a combined calibration, very limited tradeoffs in performance were apparent.*

*Demirel, M. C., Mai, J., Mendiguren, G., Koch, J., Samaniego, L. and Stisen, S.: Combining satellite data and appropriate objective functions for improved spatial pattern performance of a distributed hydrologic model, Hydrol. Earth Syst. Sci. Discuss., 1–22, doi:10.5194/hess-2017-570, 2017.*

p.2 l. 15-16: It might depend on the application of the model, sometimes the spatial pattern might even be irrelevant and a good temporal performance is sufficient. At some later point you mention that the necessary performance depends on the application of the model, but it might be useful to mention this already earlier in the introduction.

*This is an excellent comment. We will make sure to state this already in the introduction.*

p.3 l. 1-5: are the requirements for earth system models and hydrological models the same? you claim your studies findings are imporant for earth system models but all your requirements and testing seem very focussed on hydrological models.

*As mentioned earlier, we will remove the broad scope of earth system modeling and focus on hydrological modeling in the revised manuscript.*

p.3 l. 9-12: if your variable has different units, ok. but if the unit is the same you might want your model to have the same mean or at least not a large deviation. That would then require an additional metric? how would you merge it then with your multicomponent metric?

*If a bias term was desired in the spatial pattern evaluation it could easily be added to the SPAEF formulation, in a similar fashion as it is done in the KGE formulation. However we do not regard this as necessary, because bias-insensitivity allows the modeler to implicitly deal with data uncertainty. SPAEF focuses on the overall pattern and comparing the simulated and observed mean may overrate the quality of the remote sensing data. Most commonly, discharge timeseries data is available for hydrological modeling studies. Such*

*data allows for a reliable investigation of the overall water balance; i.e. the mean simulated and observed flow can be compared. However, the discharge data does not contain any information on the internal spatial variability of hydrological processes within a catchment. Here, the remote sensing data can make a significant contribution. We cannot expect that remote sensing observations can close the water balance through model calibration, but we can improve the internal distribution of hydrological variability of a catchment. Also, the remote sensing estimates represent a series of snapshots of cloudfree days in time and provide thereby not a continuous record. This further underlines why remote sensing data are not very well suited to address model biases. This will be stated clearly in the revised discussion.*

p.3 l. 15: the possibility to include data uncertainties could be another point. Remote sensing data inlude considerable uncertainties, optimizing the model by treating the "observed data" as the truth can lead to overfitting or biased model parameters especially if the uncertainties in the data scale with another important variable or increase with increasing values of the variable.

*As pointed out before, temporal aggregation and bias insensitivity are ways to implicitly deal with data uncertainty, which we will clearly state in the revised manuscript.*

p.4 l.30: this seems your way to partly deal with the data uncertainty.

*Correct.*

p.5, l.17, "source of information" this seems to be the wrong expression, probably a single metric or a single characteristic? single source of information sounds to me like using only one dataset to compare the model with as opposed to using multiple datastreams to optimize or evalute the model.

*We agree and reformulate the sentence. The term "source of information" could be changed to "single component".*

p.8, l.3: why are you doing a sensitivity analysis? Is this to select a limited set of parameters for the optimization? if yes please explain.

*Correct, we have conducted the sensitivity analysis to select a limited set of the most sensitive parameters. mHM has 48 parameters and the sensitivity analysis has identified the 17 most informative parameters which then were estimated in the calibration. The reasoning behind the applied sensitivity analysis will be clearly stated in the revised manuscript.*

p.8, l. 22-25: This seems to be a result, please move this paragraph.

*Agreed. We will move this section to the results.*

p. 12. l. 14: The insensitivity to bias can also be a disadvantage, in many cases the optimized model is desired to be unbiased.

*We totally agree to this comment. We recommend to use remote sensing data in combination with discharge timeseries for the calibration of spatially distributed models. The discharge data will ensure that the overall waterbalance is in place (i.e. unbiased) and the remote sensing data will constrain the catchment internal distribution of fluxes. Again, the remote*

*sensing data is only obtained at cloudfree days and therefore does not provide a full record. This hampers the suitability of remote sensing data to assess model biases. We will clearly point this out in the revised manuscript. Especially that the bias insensitivity in the SPAEF metric is only reasonable when being accompanied by discharge data.*

p.12, l. 15: if the units differ, it might depend how the two units relate to each to other. it certainly is ok if they linearly scale. How about a nonlinear relationship? How about a possible change in sign as for instance with celsius and kelvin? if the mean temperature in celsius would go towards zero you would get difficulties for the beta part of your metric?

*This is a very interesting point which will be discussed in the revised manuscript. We will advise the reader to investigate the relationship between the variables to be compared by SPAEF. In case there is a non-linear relationship one may consider to log transform the data. The variability term in the SPAEF formulation is mean normalized which should be quite robust. The histogram term is based on the z-score transform of the data which should also work for most cases. However we will inform the reader about alternative ways of normalization which may be relevant for certain cases. In case of non-linear relationships the transformation could be especially relevant for the correlation coefficient which assumes linearity.*

Reproduceability: Will you provide your model outputs, observations used and analysis scripts?

*All scripts used in this study are made available via GitHub and citable via Zenodo. Model outputs and observations will be made available upon request.*
*https://github.com/cuneyd/spaef*
*https://github.com/JulKoch/SEEM*
*https://github.com/mhm-ufz/mhm*